# An Effective Method for Detection and Recognition of Uyghur Texts in Images with Backgrounds

**Mayire Ibrayim, Ahmatjan Mattohti and Askar Hamdulla ***

College of Information Science and Engineering, Xinjiang University, Urumqi 830046, China;
mayire401@xju.edu.cn (M.I.); sayyarim@163.com (A.M.)
* Correspondence: askar@xju.edu.cn; Tel.: +86-139-9922-1222

**Abstract:** Uyghur text detection and recognition in images with simple backgrounds is still a challenging task for Uyghur image content analysis. In this paper, we propose a new effective Uyghur text detection method based on channel-enhanced MSERs and the CNN classification model. In order to extract more complete text components, a new text candidate region extraction algorithm is put forward, which is based on the channel-enhanced MSERs according to the characteristics of Uyghur text. In order to effectively prune the non-text regions, we design a CNN classification network according to the LeNet-5, which gains the description characteristics automatically and avoids the tedious and low efficiency artificial characteristic extraction work. For Uyghur text recognition in images, we improved the traditional CRNN network, and to verify its effectiveness, the networks trained on a synthetic dataset and evaluated on the text recognition datasets. The experimental results indicated that the Uyghur text detection method in this paper is robust and applicable, and the recognition result by improvedCRNN was better than the original CRNN network.

**Keywords:** text detection; text recognition; channel enhanced MSERs; CNN; CRNN

## 1. Introduction

Scene text detection is an important research direction in the field of object detection, with the goal of detecting text in real-world photographs. Recently, with the upsurge of deep learning, scene text detection has been widely applied in driverless, unmanned supermarkets, image retrieval, real-time translation, and other fields.

Text is one of the most important ways we perceive our surroundings, and as one of humanity's most brilliant and influential creations, it has played an important role in modern society [1–3]. As a carrier of information exchange between people, text is highly abstract and widely exists in various images. However, the text in the image has a stronger logic and generality, and it can more effectively provide accurate high-level language description and rich semantic information, which is helpful in analyzing and understanding the content of the image.

Although text detection and recognition in images with complex backgrounds has achieved great success in the majority of languages, such as English and Chinese, research on Uyghur text detection and recognition in images with simple backgrounds has remained a difficult challenge in recent years. The challenge mainly comes from the peculiarities of Uyghur text, which is very different from majority language characters is several aspects, as illustrated in Figure 1:

1.  The Uyghur text is composed of a main part and an additional part at the image level. The characters in the main part always stick to each other, which belongs to a connected domain. The additional part is secondary but should not be ignored, which has one to three dots or other special structures.
2.  Words are the basic units of Uyghur text, and the length of each word is different.

3. There are no uppercase or lowercase letters, but each letter has a different form, and use different forms in different locations.
4. There is a "baseline" for every sentence in the middle of the text. Usually, each word on the baseline has the same height and the distance between them is similar.

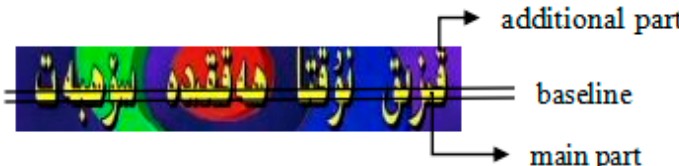

**Figure 1.** The characteristics of Uyghur text.

To detect the Uyghur text in images with simple backgrounds, the Maximally Stable Extremal Regions (MSERs) [4] algorithm is always chosen for its good performance. But the traditional MSERs methods extract MSER regions on grayscale images. When transforming the color images to grayscale, the contrast between objects and background usually becomes weaker, and this leads to some fuzzy regions being missed. Therefore, the channel-enhanced MSERs algorithm was proposed, which uses the R, G and B channels of the image to be processed, and then the MSER regions on the three channels are marked on the color image. But the channel-enhanced MSERs algorithm is not very good for detecting Uyghur text completely.

In this paper, we proposed an effective and applicable Uyghur text detection method based on the channel-enhanced MSERs [5] and CNN classification model, which is designed with the characteristics of Uyghur text in images with a simple background. In the component extraction stage, a new component candidate extraction algorithm is put forward, which is based on the channel-enhanced MSERs [5] according to the characteristics of Uyghur text. In component analysis stage, the CNN classification model replaces the SVM classifier, which is trained with the HOG feature of the components. And a merging candidate regions algorithm is proposed to build the word-level text candidate regions. Recall the omitted text regions according to the color similarity and spatial relationship of the word-level text candidate regions, classify them with CNN, and merge the text regions into text lines at last.

For text recognition, traditional methods usually process individual characters first and then integrate them into words by beam search, dynamic programming, etc. [6]. The recent methods cast the text recognition task as a sequence recognition problem. Most of the methods are used to recognize majority languages, such as English and Chinese [7,8]. In order to recognize Uyghur text in images with a simple background, in this paper the Convolutional Recurrent Neural Network (CRNN) was improved, and then carried on the text sequence recognition with the improvement of the CRNN model to the Uygur texts in images. In this paper, the Uyghur texts in images were labeled in Latin, and then the decoding results of the transcription layer were converted into Latin characters. The Latin characters were converted into Uyghur text sequences finally.

The main contributions of this paper are as follows:

1. In view of the Uygur text characteristic, a new effective Uyghur text detection method based on channel enhanced MSERs and CNN is proposed.
2. We improved the traditional CRNN network, and then carried on the text sequence recognition with the improved CRNN model to the Uygur texts in images with a simple background.
3. In order to satisfy the image sample data needs for the text detection duty, the text detection data set was established. In order to meet the training and testing requirements of the improved CRNN network, a random text in the natural scene image production tool [9] was developed independently, and the text recognition dataset was established.

The remaining sections are organized as follows. In Section 1, The format of the Uyghur language is briefly described, and the main points of contribution of this paper are generally presented. In Section 2, some related works about Uyghur text detection and recognition in images are briefly reviewed. In Section 3, the proposed methodology is described in detail. In Section 4, the results of the conducted experiment are illustrated. And in Section 5, the conclusion and discussion of the paper are demonstrated.

## 2. Related Work

Optical Character Recognition (OCR) research has a long history, and text detection and recognition in images has achieved enormous success in English and Chinese [10–14]. But the characters of Uyghur text are quite different from the majority languages, so we need to improve the text detection and recognition methods according to the characteristics of Uyghur text.

Song et al. [5] generated text components based on a multi-color-channel enhanced MSERs algorithm to detect the Uyghur text in the complex background images precisely. According to the strong baseline characteristic of Uyghur text, the HOG descriptor was applied to represent the Uyghur text. Then they employed a powerful SVM classifier to identify the text components. However, the method they proposed was unable to extract complete Uyghur text from candidate regions. Besides, it works at low efficiency.

Fang et al. [15] developed an effective and efficient region-based convolutional neural network for Uyghur text detection in complex background images. Their system took an image as input and output the confidence and locations of the predicted text boxes. The feature extraction network first extracted the CNN features from the image. Then three region proposal networks generated text region proposals from three different convolutional layers. Finally, the text detection network confirmed the proposals. The whole network was trained to detect text in an end-to-end fashion.

For text recognition in images, there are two main approaches: bottom-up and top-down. The typical bottom-up methods first detect characters one by one with a series of operations, then recognize recognize the single character one by one, and then combine them into the final text. Alsharif et al. [16] achieve the recognition of English words of any length through the combination of two CNN models and a hybrid HMM Maxout model. First the CNN network divides the text in the natural scene image into a large number of segments, then classifies each segment, and finally the hybrid HMM Maxout model is used to sort out the classification results. The top-down methods make predictions directly on the original image without detecting the single character firstly. Jaderberg et al. [17] used a deep convolutional neural network to perform recognition tasks, including unconstrained recognition and a 90k-class classification task for English words. This method does not need to detect and recognize every single character one by one, and recognizes the words as a whole. But the method is unable to recognize words or numbers of any length that are not in the lexicon.

Recent research regarded the recognition task of textual images as a sequence identification problem and solved it by using RNN. Shi et al. [18] presented the Convolutional Recurrent Neural Network (CRNN), a novel neural network architecture that integrates the advantages of both Deep Convolutional Neural Networks and Recurrent Neural Networks. The network architecture of CRNN has three components, including the convolutional layers, the recurrent layers, and a transcription layer, from bottom to top. The convolutional layers extract a feature sequence from the input image; the recurrent layers predict a label distribution for each frame, and the transcription layer translates the per-frame predictions into the final label sequence. The CRNN is able to take input images of varying dimensions and produce predictions of varying lengths. It directly runs on coarse level labels, requiring detailed annotations for each individual element in the training phase.

The mentioned methods focus on recognizing English, Chinese and numbers. However, in the scenario of Uyghur text recognition in images, the types of characters are not only Uyghur language and numbers but also Chinese, English, and other special symbols.

The character types and forms, especially the complex Uyghur character structure, make it more difficult to recognize correctly. In order to address these challenges, this paper improved the CRNN network and used the improvement of the CRNN network to recognize the Uyghur text in images.

### 3. Methods

In this section, first we elaborate the text detection method based on channel-enhanced MSERs and CNN, which is designed with the merits of channel-enhanced MSERs-based methods and CNN classification model. Then we introduced the CRNN and how to improve the network effectively, and then carried on the text sequence recognition with the improved CRNN network to the Uygur text in images.

#### 3.1. Text Detection Method Based on Channel-Enhanced MSERs

The text detection method based on channel-enhanced MSERs and CNN consists five stages, as illustrated in Figure 2. Firstly, we dilate the R, G and B channels of the image to be processed, then extract candidate regions using the channel-enhanced MSERs algorithm, and filter the candidate regions according to the heuristic rules of text features. Then, a CNN classification model is used to classify the candidate regions and reserve the text regions, and the word-level text candidate regions are built by merge candidate regions algorithm. Recall the omitted text regions according to the color similarity and spatial relationship of the word-level text candidate regions. Then classify the recalled regions with CNN and merge the text regions into the word-level text candidate regions. Finally, link the word-level text candidate regions according to the spatial characteristics of the text to localize the image text regions.

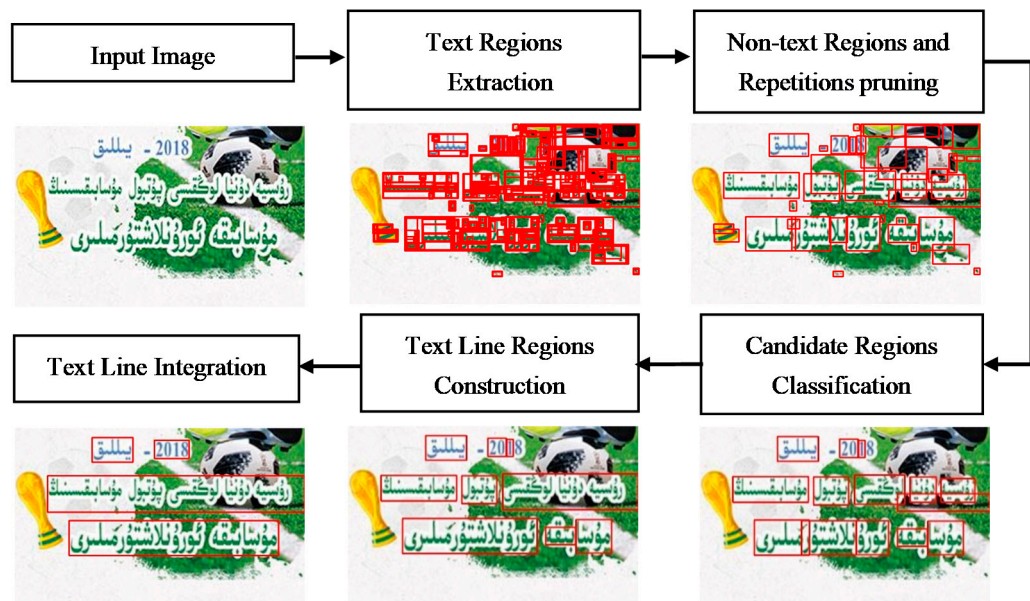

**Figure 2.** Flowchart of the proposed method and corresponding results of each step.

(1) Text Regions Extraction: at this stage, a new text candidate region extraction algorithm is put forward, which is based on the channel-enhanced MSERs according to the characteristics of text. Firstly, dynamically adjust the size of the input image, because it will cost more time if the input image is too large size and small text candidate regions missed if the input image is too small size. Then, the image is separated by channels and the B, G and R single-channel image is dilated respectively. In this way, the additional parts of the text in images are added to the main parts as much as possible to form a connected area. The MSERs [5] algorithm is executed in B, G and R single-channel respectively, then all candidate regions are marked up in the input

image. We can see that almost all text candidate regions can be extracted using above method. But there are more overlapping regions and obvious non-text regions, so we have to remove them quickly.

(2) Non-text Regions and Repetitions pruning: there are many regions returned in the Text Regions Extraction stage and most of them are overlapping regions and obvious non-text regions, which should be pruned. The obvious overlapping regions and non-text regions are pruned by setting some simple heuristic rules. Firstly, completely overlapping regions are pruned according to the position coordinates, and then the areas that are too large or small are pruned by calculating the area of the regions. If there are two not completely overlapping regions, if the intersection area is over 80 percent of the larger region, the smaller is pruned. In this way, most overlapping regions and obvious non-text regions are pruned.

(3) Candidate Regions Classification: most of overlapping regions and obvious non-text regions are pruned by simple heuristic rules, but it is still necessary to determine the candidate regions are belong to text or not. In order to quickly and effectively remove the non-text regions which are hard to be pruned by heuristic rules, a simple CNN classification network is designed in reference to AlexNet [19]. The CNN classification model is trained with the crop image of the candidate regions. In terms of the characteristics of words and the phenomenon of many experimental comparisons, the crop image samples are all resized to $48 \times 32$ (*width* $\times$ *height*) pixels (the results with different size will be discussed in the experiments). The most of the text regions are determined by the Candidate Regions Classification, and the non-text regions are removed from the candidate regions.

(4) Text Line Regions Construction: after Candidate Regions Classification, only the text regions and a small number of non-text regions are left. We can see that some of the words are marked by multiple candidate areas, so that each candidate region is a part of the whole word and is unable to mark the complete vocabulary. For text detection, the candidate regions are connected into the word-level text candidate regions, which are built by the merge candidate regions algorithm. Firstly, the regions are connected together, which have intersection, and this process is taken from the left of the image to the right and from top to bottom. Then, if the spatial distance between the center positions of the two candidate regions is smaller than the distance between words, the regions are connected. A new rectangle is computed to contain the two regions when connecting two candidate regions.

(5) Text Line Integration: one or more word-level text candidate regions appear on one horizontal line after Text Line Regions Construction, and a few text regions are omitted by the above stages, because the channel-enhanced MSERs algorithm is unable to detect all text regions, some of text regions are removed by the heuristic rules as non-text regions; the simple CNN classification model misclassifies the candidate regions and so on. In this stage, every word-level text candidate region sequentially searches forward and backward according to the color similarity and spatial relationship of the word-level text candidate regions. In this way, many regions are recalled as text regions, then they use the above CNN classification model to classify the recalling regions, and aggregate the omitted text regions into the word-level text candidate regions. Finally, connect the all regions together, and the text line is the final text detection result.

### 3.2. Uyghur Text Recognition Method Based on the Improved CRNN Network

The text recognition model based on CRNN is an excellent model in the field of sequence recognition, and is widely used in many texts' recognition tasks, such as the text recognition tasks in English and Chinese. The text is a sticky language text, words are basic unit of text lines and the way of writing is very special. But the text is written from right to left on the baseline in the sequence of appearance, and the CRNN model is just good at dealing with this orderly written text sequence. Thus, in this paper, the CRNN model is

used for text recognition in images. The network architecture of text recognition is based on improvement of CRNN shown in Figure 3.

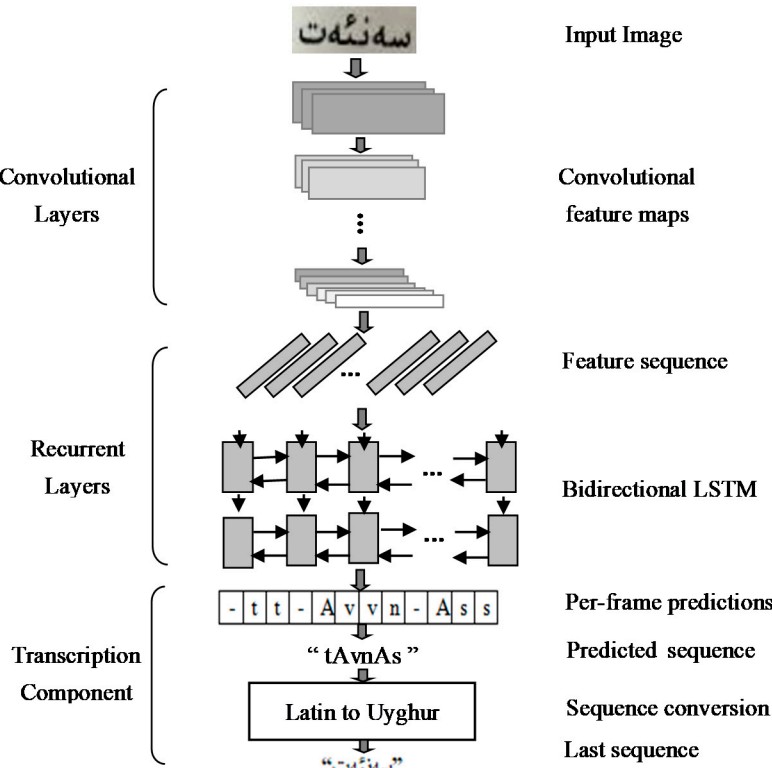

**Figure 3.** The network architecture of Uyghur text recognition based on improvement CRNN.

At the bottom of improvement CRNN is the convolutional layers, which automatically extract a feature sequence from each input image. On top of the convolutional network, a recurrent network is built for making predictions for each frame of the feature sequence, which is outputted by the convolutional layers. The transcription Component at the top of improvement CRNN is adopted to translate the per-frame predictions by the recurrent layers into a label sequence, then the Latin sequence is converted into a text sequence by sequence conversion. The improvement CRNN is composed of different kinds of network architectures, but it can be jointly trained with one loss function.

The CRNN network is designed to recognize English initially, so the third and fourth Max Pooling Layers in the network use the pooling step instead of the pooling step, such pooling step bring a narrow Receptive Field, which makes it easier to distinguish narrow-width letters, such as i, j, l and so on. The text is a sticky language text, and each word is written continuously and each individual letter is generally similar in length and width. At the same time, there are many relatively long words. Therefore, the feature sequence is long enough to recognize the correct sequence of letters. The CRNN network requires that the height of the input images is 32 pixels. Generally, the width of the input images of Uyghur words is 192 pixels, when the height is 32 pixels. When the size of input image is, the CRNN network obtains a feature sequence with 48 characters, which is too long. If the third and fourth Max Pooling Layers in the network use the pooling step, the CRNN network obtains a feature sequence with 12 characters, which is too short. Thus, in this paper, the 3rd and 4th Max Pooling Layers in the network use the and pooling step, respectively. In this way, the network obtains a feature sequence with 24 characters, which is suitable for most Uyghur texts. Since we have limited training data, the 4th and 6th convolution layers removed in order to prevent the over-fitting and reduce the capacity of the model.

It is a huge project to label the images in a data set, which takes a lot of time. Thus, it is necessary to automatically label the image in the dataset. The random text in the natural

scene images production tool synthesizes a large number of word images and obtains the corresponding label sequence together. When labeling the word images in Uyghur, the letters in prediction sequence are unable to match the labeling sequence very well, because each letter in the Uyghur language has several forms and the current experimental platforms do not handle Uyghur text very well, which makes it hard to train the model effectively. So, in this paper, the Uyghur texts in images were labeled in Latin, and the correspondence between them is shown in Figure 4.

**Figure 4.** The correspondence of Uyghur language and Latin.

From the above correspondence of the Uyghur language and Latin, we can see that the 33 Latin letters replace all forms of Uyghur language letters. At the same time there are three special punctuation marks in the Uyghur language that replace normal marks correspondingly. So, we denote the Uyghur language alphabet set by $U$, which contains 33 letters and 3 special marks. The CTC layer at the top of the CRNN network plus a 'black' label denoted by "-". Therefore, the label set of the CTC layer is denoted by $V = U \cup \{-\}$, which contains 37 labels.

## 4. Experiment

In this section, new datasets are built to train and test the proposed methods, which contains images with simple background. Each image may contain one or more Uyghur text lines, and the backgrounds of the images are easy to confuse with the texts, so it challenging for text detection and recognition. All of these experiments are evaluated on these new datasets.

### 4.1. Datasets

We collected each type of text in the natural scene image and classified it, reorganized and divided it, resulting in the text detection data set, which contains 386 color images, as shown in Figure 5a. The training set consists of 266 randomly selected text in the natural scene images, which are used to construct a cropped regions dataset for training and testing the CNN classification model. The cropped regions dataset contains 16,683 cropped regions that are cropped from text in the natural scene images, with the positive samples consisting of 8485 cropped text regions (examples shown in Figure 5b) and the negative samples consisting of 8398 cropped text regions (examples shown in Figure 5c. The testing set consists of 120 randomly selected text in the natural scene images, which are used to evaluate the whole text detection method.

In order to meet the training and testing of the improvement of CRNN network, a random text in the natural scene image production tool was developed independently, and a random synthetic texts image dataset was established, which contains more than one million fixed size text in the natural scene images, examples are shown in Figure 6a and the detail are shown in Table 1, and an arbitrary length text in the natural scene image dataset was established, which contains 102,960 arbitrary length text images. Examples are shown in Figure 6b.

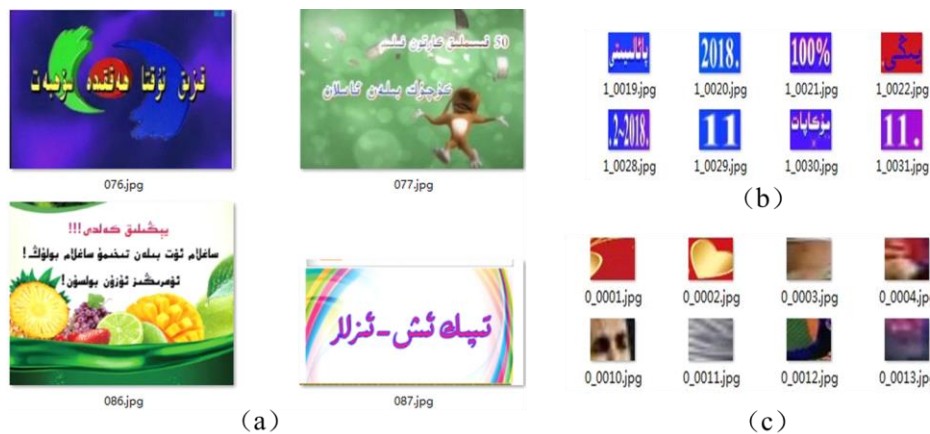

**Figure 5.** Examples of the detection dataset. (**a**) examples of color images; (**b**) positive samples in training set; (**c**) negative samples in training set.

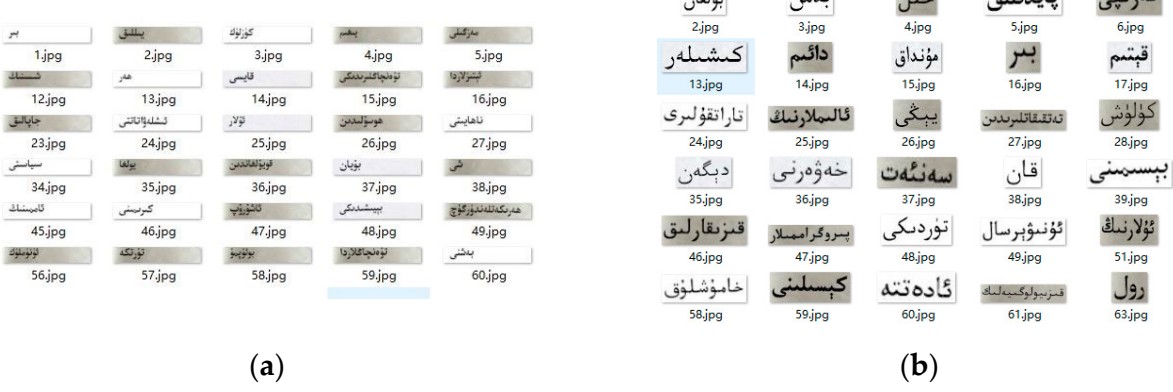

(**a**)                                    (**b**)

**Figure 6.** Examples of the recognition dataset. (**a**) examples of random synthetic texts images; (**b**) examples of arbitrary length text in the natural scene images.

**Table 1.** The information of random synthetic texts images data set.

|        | Training Set | Validation Set | Testing Set | Total     |
|--------|--------------|----------------|-------------|-----------|
| Images | 823,680      | 102,960        | 102,960     | 1,029,600 |
| words  | 68,640       | 8580           | 8580        | 85,800    |

### 4.2. Performance Evaluation Criteria

In order to quickly and effectively remove the non-text regions which are hard to be prune by heuristic rules, a simple CNN classification model is designed with reference to AlexNet, and the network architecture show in Figure 7. The input images are all resized to $48 \times 32$ (*width* $\times$ *height*) pixels, and the result of the output is the label of text or non-text. The network is trained with the Adam optimization algorithm. The learning rate is set to 0.001 and the multiplier factor (gamma) of the learning rate is set to 0.1, and the loss function select the Cross Entropy Loss [20]. And the CNN classification model trained by the above cropped regions dataset, which contains positive samples and negative samples. When doing training, the samples are resized to the same size. As we only focus on Uyghur text detection, the CNN classification model is only trained on Uyghur texts and numbers.

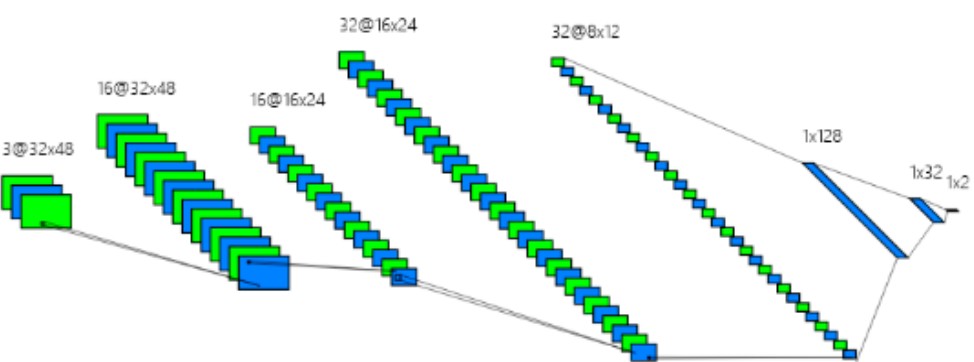

**Figure 7.** The network architecture of CNN classification model.

The performance of the CNN classification model can affect the result of the detection method directly, if the classifier could prune as many non-text regions as possible, the precision of the system would be higher. But in order to train the CNN classification model effectively, the input images are resized into the same size. However, if the size is set too large or too small, it always loses text information. So in the experiment, the size of the region samples is resized to $32 \times 24$, $32 \times 32$, $48 \times 32$, $48 \times 48$ and $64 \times 48$ (*width* $\times$ *height*). And the accuracy and loss of the CNN classification model with different sizes of the region samples are different. The details of the change are shown in Figure 8. As illustrated in Figure 7, the average loss of the CNN classification model is almost same in different size, but the CNN classification model has the highest accuracy value when the regions are set to $48 \times 32$.

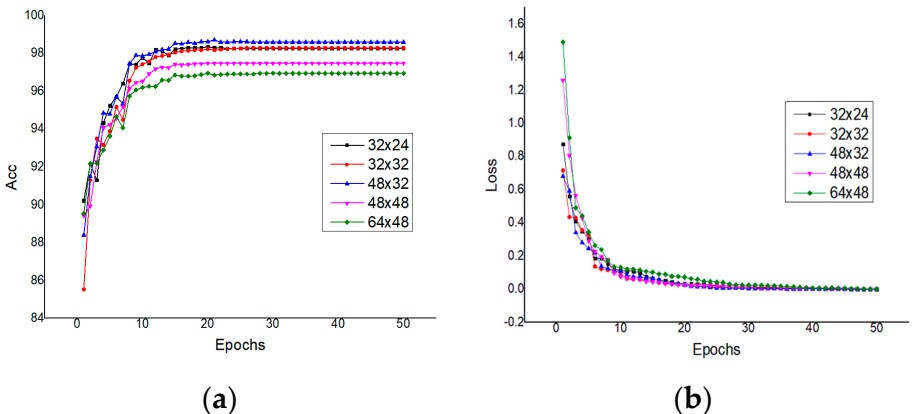

|             (**a**)             |             (**b**)             |

**Figure 8.** The performance of the CNN classification model in the Validation set. Subfigure (**a**) the accuracy changing curve of the CNN classification model; Subfigure (**b**) the average loss changing curve of the CNN classification model.

*4.3. Experiments on Uyghur Text Detection*

4.3.1. Implementation Details

In this paper, experiments are conducted on the relationship between the training method and the size of sample data used for training and testing and the prediction performance of CNN networks. The experimental platform is a Mac laptop with Intel Core i5 CPU and 3.1 GHz main frequency, and the deep learning tool is PyTorch. the CNN network model in this paper is a network structure with a depth of 5 layers designed with reference to AlexNet. When importing data, the order of the imported data is firstly disordered and the data are randomly divided in the ratio of 8:2, which are used for training and testing, then the input images are normalized and the imported images are transformed using normalization parameters with means of 0.485, 0.456, 0.406 and variances of 0.229, 0.224, 0.225. For training, the cross-entropy loss function is used for the loss function, and

the optimization method uses Adam, a first-order optimization algorithm that can replace the traditional stochastic gradient descent process.

### 4.3.2. Comparisons with Existing Methods

To mark the text lines on one image, a rectangle around every detected text should be drawn out. However, it is a problem that how to judge whether a text line is correctly detected. For the estimated rectangle *D* and the ground truth rectangle *G*, the overlap ratio is calculated to judge whether *D* is correctly detected. The overlap between *D* and *G* is defined as:

$$r(G, D) = \frac{area(G \cap D)}{area(G \cup D)} \tag{1}$$

where $area(G \cap D)$ and $area(G \cup D)$ represent the areas of the intersection and union of *G* and *D*. If the overlap ratio is larger than 0.5, the rectangle is considered as a correct detection.

The precision (*P*), recall (*R*) and F-measure are used to evaluate the text detection method, and the definitions of them are as follows:

$$p = |T_p|/|E| \tag{2}$$

$$R = |T_p|/|T| \tag{3}$$

$$R = 2PR/(P + R) \tag{4}$$

where $T_p$ is the true positive detection set, *E* and *T* are sets of estimated rectangles and ground truth rectangles, respectively.

The performance is assessed by the above evaluation protocol, and to verify the performance of our algorithm, we compared our method with text detection methods based on the MSERS and CNN, and the results are shown in Table 2. From Table 2, we can see that the precision, recall and F-measure of our method all have higher values. The traditional MSERs detection method extracts candidate regions from the grayscale image, and when transforming the color images to grayscale, the contrast between objects and background usually becomes weaker, which leads to some fuzzy regions missed. The channel-enhanced MSERs method is used in the R, G and B channels of the image to be processed, which makes full use of color information and fetchs up the fuzzy regions missing. Our method is designed with the characteristics of Uyghur text in images with a simple background and makes full use of the advantages of channel-enhanced MSERs method. So our method has a good performance. The Precision, Recall and F-measure on the test data have reached 0.881, 0.872 and 0.876 respectively. The experimental results indicated that the algorithm in this article is robust and applicable.

**Table 2.** The evaluation results in uyghur text detection.

| Methods | P | R | F |
|---|---|---|---|
| Our method | 0.881 | 0.872 | 0.876 |
| Channel-enhanced MSERs + CNN | 0.846 | 0.835 | 0.840 |
| Traditional MSER + CNN | 0.812 | 0.804 | 0.808 |

### 4.4. Experiments on Uyghur Text Recognition

### 4.4.1. Implementation Details

All experiments mentioned in this paper use pytorch as the deep learning framework, the programming compiler is python 3.6, the initial learning rate is set to 0.0001, the batch size of the training set is 128, and the test set is 64. We train the model from scratch using the Adadelta optimizer, and the training period is set to 30 epochs. We use 823,680 synthetic images as the training set, 102,960 samples in the test set, and the input images are resized to $32 \times 184$ and then fed into the neural network for training.

In order to make the model converge faster, we borrowed the idea of migration learning for this experiment. Specifically, Synthetic Word Dataset was used for pre-training,

and then the Uyghur dataset labeled in Latin form was used for fine-tuning, during which the 10 model weights with the best training performance were saved.

Unlike the training, the test session fixes the height of the input image to 32 pixels and scales the width equally, and then feeds it into the improved CRNN model to obtain the predicted sequence encoding and transcription decoding, and finally obtains the corresponding Latin characters. Since the final recognition results need to be presented in Uyghur, the obtained Latin content is converted into Uyghur textual content using the Latin to Uyghur conversion relation.

4.4.2. Comparisons with Original Method

In order to objectively evaluate the text recognition performance of the improved CRNN network model, the accuracy of prediction and Average Edit Distance (*AED*) is used to evaluate:

$$accuracy = \frac{m}{N} \tag{5}$$

$$AED = \frac{\sum_{i=1}^{N} ED(x_i^p, x_i^r)}{N} \tag{6}$$

where $N$ and $m$ represent the number of all test images and correct recognition, respectively, $x_i^p$ and $x_i^r$ are the recognition and ground-truth of the $i$-th image, respectively. For one test image, the result is considered correctly if and only if all the characters in the prediction are the same as ground truth without repetition and omission.

The CRNN and the improvement CRNN are trained by the random synthetic text images dataset, and then two of them are tested with the testing set of the random synthetic text images dataset and the arbitrary length text in the natural scene images dataset, respectively. Firstly, the testing set in the random synthetic texts images dataset is used to evaluate the CRNN and the improved CRNN network respectively, and the test results shown in Table 3. From Table 3 we can see that the improved CRNN network in this paper has increased the accuracy and decreased the AED. The result shows that the improved method based on the CRNN network, which is given in this paper, is able to recognize the texts in images.

**Table 3.** The evaluation results in uyghur text recognition.

| Networks | Accuracy | AED |
| --- | --- | --- |
| The Original CRNN | 0.782 | 0.458 |
| The Improved CRNN | 0.864 | 0.204 |

From Table 4, we can see that both network models can recognize most text in images, but a large number of false and repeated predictions occur in the output text by the original CRNN. The receptive field of the last convolutional layer in the original CRNN is relatively narrow, and then it is hard to recognize the similar characters. For example, the characters " ☸ " and " ☸ " are similar, the original CRNN recognizes " ☸ " as " ☸ ", but the improved CRNN recognize it correctly. The improved CRNN use a wider receptive field which is closer to the width of Uyghur characters, so it is more suitable for Uyghur text recognition, but false and repeated predictions still occur in the output text. For example, the receptive field of the feature extracted by CNN divides the character " ﺈﺌ " into " ﺌ " and " ﺍ ", and the two parts are recognized as " ﺈﺌ " and " ﺍ ", in this case, the two networks recognize the additional chapter " ﺍ ".

**Table 4.** Experimental results at 640 × 640 scale on the Total-Text dataset, Pre-Train represents whether to use the pre-training dataset for fine-tuning. "Y" means using pre-training, "N" means not using pre-training. "P", "R" and "F" stand for "precision", "recall" and "F-measure" respectively.

| Input Images | Networks | Results in Latin | Results in Uyghur | ED | Time |
|---|---|---|---|---|---|
| ئىستاتىستىكىلىق | The Original CRNN | vistatistikiliq | ئىستاتىستىكىلىق | 0 | 0.057 |
| | The Improved CRNN | vistatistikiliq | ئىستاتىستىكىلىق | 0 | 0.039 |
| سۇتچىلمك | The Original CRNN | sUttcilikk | سۇتتچىلمكك | 2 | 0.062 |
| | The Improved CRNN | sUtcilik | سۇ تچىلمك | 0 | 0.047 |
| مۇگىدىشدىغنىمىزغا | The Original CRNN | muNdixidiHinimizGa | مۇگىدىشد مخنمىمزغا | 1 | 0.094 |
| | The Improved CRNN | muNdixidiGinimizGa | مۇگىدىشد بغنىمىزغا | 0 | 0.062 |
| چىپنىقتۇرۇشنلڭ | The Original CRNN | cpeniqturuxinilN | چىپنىقتۇرۇشنلڭ | 3 | 0.062 |
| | The Improved CRNN | ceniqturuxnilN | چىپنىقتۇرۇشنلڭ | 1 | 0.047 |

In order to verify the performance of the CRNN network and the improved CRNN network, the arbitrary length text in the natural scene image dataset is used to evaluate them, respectively, and the test results are shown in Table 5. From Table 5, we can see that the improved CRNN network in this paper has improved the accuracy and AED, but the improvement is not very obvious. The accuracy is low and the average editing distance is also large, and it is hard to use in practical applications. At the same time, the accuracy and AED become worse than the results in Table 3, because the arbitrary length text in the natural scene images have to resize to the $w \times 32$ (*width* × *height*) pixels, which makes some distortion, deformation and missing information for the input images. The examples of recognition cases are shown in Table 5.

**Table 5.** The evaluation results in uyghur text recognition.

| Networks | Accuracy | AED |
|---|---|---|
| The Original CRNN | 0.728 | 0.716 |
| The Improved CRNN | 0.780 | 0.492 |

From Table 5 we can see that the both network models can recognize most text in image, but the false and repeated predictions occur obviously in the output text. The the improvement of CRNN network is better than the original CRNN for Uyghur texts recognition in the arbitrary length text in the natural scene images, but it is worse for

recognizing the shorter words. For example, the improved CRNN recognizes " ئۇ " as " ۇ ", but the original CRNN recognize it correctly. The improvement of CRNN network also has the problem of false and repeated predictions, but the recognition result is closer to the correct text sequence, and the editing distance and running time of testing are less than the original CRNN network. In general, the performance of the improved CRNN network in this paper is better than the original CRNN in the arbitrary length text in the natural scene image dataset, but its accuracy needs to be further improved.

From the recognition results in Table 6, we can see that both network models are able to recognize most of the word image text, and the recognition results of the improved CRNN network model are closer to the correct text sequence i.e., the editing distance is smaller than that of the original CRNN network model, and also the running time of the algorithm is smaller than that of the CRNN network model. Overall, the improved CRNN network algorithm in this paper performs Better than the original CRNN model on word image dataset of arbitrary length, but its accuracy needs further improvement.

**Table 6.** The evaluation results using different methods.

| Input Images | Networks | Results in Latin | Results in Uyghur | ED | Time |
|---|---|---|---|---|---|
| كەتكەندىن | The Original CRNN | kAtkAndin | كەتكەندىن | 0 | 0.0468 |
|  | The Improved CRNN | kAtkAndin | كەتكەندىن | 0 | 0.0311 |
| ئۇ | The Original CRNN | vu | ئۇ | 0 | 0.0156 |
|  | The Improved CRNN | u | ۇ | 1 | 0.0156 |
| سىستېمىسىدا | The Original CRNN | sistemissida | سىستېمىسسىدا | 1 | 0.0468 |
|  | The Improved CRNN | sistemisida | سىستېمىسىدا | 0 | 0.0312 |
| ھەيدىشكۈزنى | The Original CRNN | Aydixiizni | ەيدىشىزنى | 2 | 0.0468 |
|  | The Improved CRNN | hAydixiNizn | ھەيدىشكۈزن | 1 | 0.0312 |

## 5. Conclusions and Future Work

In this paper, an effective Uyghur text detection method in images with simple backgrounds based on channel enhanced MSERs and the CNN model is proposed. In order to satisfy the image sample data needs for in the text detection duty, the text detection data set was established. And the experimental results indicated that the Uyghur text detection method is robust and applicable. Then the CRNN model improved, and carried on the text sequence recognition with the improved CRNN model to the Uygur texts in images. In order to meet the training and testing requirements of the improved CRNN network, a random text in the natural scene image production tool was developed independently, and the text recognition data set was established. The accuracy and AED of the improved CRNN are better than the original network, and the network is able to recognize the text in images.

The major contribution of this study is that it puts forward a flexible way to detect text in images with simple backgrounds and suggests a new recognition idea in Uyghur text based on deep neural networks.

The Uyghur text detection method based on the improved CRNN network does not improve the encoding and decoding part of the network. And use Latin to label the Uyghur text in the natural scene images. In this way avoid the problems that occurred in the encoding and decoding of sticky words, but it does not fundamentally solve the problem. So the encoding and decoding parts of the CRNN network model need to be improved in future work. Besides, in this paper we only focus on the image text recognition that contains only Uyghur words, but the text in images also contains punctuation, numbers, English, and even Chinese characters. Therefore, the recognition of mixed text in images and scene text needs further study in the future.

**Author Contributions:** Conceptualization, M.I.; methodology, M.I.; software, M.I., A.M. and A.H.; validation, M.I. and A.M. and A.H.; formal analysis, M.I.; investigation, M.I., A.M. and A.H.; resources, M.I.; data curation, M.I.; writing—original draft preparation, M.I.; writing—review and editing, M.I. and A.H.; visuali-zation, M.I.; supervision, A.H.; project administration, A.H.; funding acquisition, M.I. and A.H. All authors have read and agreed to the published version of the manuscript.

**Funding:** This work is supported by Natural Science Foundation of Xinjiang (No. 2020D01C045), National Science Foundation of China (NSFC) under Grant No. 62166043 and Youth Fund for scientific research program of Autonomous Region (XJEDU2019Y007).

**Institutional Review Board Statement:** Not applicable.

**Informed Consent Statement:** Not applicable.

**Data Availability Statement:** Data used to support the work of this study are available from the corresponding author upon request.

**Conflicts of Interest:** The authors declare no conflict of interest.

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
