# Peer review of "An Effective Method for Detection and Recognition of Uyghur Texts in Images with Backgrounds"

_information, doi:10.3390/info13070332_

Round 1

Reviewer 1 Report

Overall, I am quite satisfied with the technical quality of this submission. The problem is a bit niche but otherwise relevant and I am sure of interest to a specific sub-community; the contribution is sufficient, and the experiments are suitably convincing. 

Having said the above, the quality of writing must be improved significantly before this submission can be fully accepted. Just a small number out of many examples:

  • "text in natural scene": missing article.

  • "and the multi-layer features": article not needed.

  • "As for text recognition in images, there are already two major methods": very poor phrasing.

  • "The one CNN": again unnecessary article.

  • "Firstly, dilate the R, G and B channels of the image to be processed, then extract candidate regions using the channel-enhanced MSERs algorithm, and the candidate regions are": poorly structured sentence which starts with imperative just to abandon it later.

  • "Error! Reference source not found..": Latex issue

  • "the result with different size have been discussed": wrong tense.

  • "the improvement CRNN network": noun/adjective confusion.

  • "obvious advantages": inappropriate.

  • "Our methods" (in Table 2): plural instead of singular

Etc., etc.

Reviewer 2 Report

The current work proposes a new effectively Uyghur text detection method based on channel-enhanced MSERs and the CNN classification model.

The problem is well explained and framed. Also, the aims and contributions of the work are well established. The related work section is up-to-date, and criticism is applied to the analysis performed.

Please consider some concerns:
(1) The major drawback is in writing. The manuscript needs an in-depth review regarding grammar and sentence construction. The are many sentences without a verb. This concern must be taken very seriously by the authors.
(2) The authors should check the references. For example, reference 9 is incomplete (no publication year is provided).
(3) Image quality in some figures should be improved.
(4) Section 4 needs further information. I don't believe the results can be replicated, and the training stage is marginally addressed.

Round 2

Reviewer 1 Report

My comments, mostly of cosmetic nature, have been adequately addressed.

Author Response

Thanks for your suggestion.

Reviewer 2 Report

Almost all concerns were adequately addressed, but unfortunately, with one exception. I still believe that even an experimented reader cannot replicate the results presented here. I invite the authors to reproduce the results and annotate the algorithms and their parametrization. Due to this fact, I keep my recommendation for a major review.

My apologies for the insistence but consider this aspect of utmost importance.

Round 3

Reviewer 2 Report

The authors adequately answered the questions raised.